# Exploration of the Shoulder Internal Rotation’s Influence on Throwing Velocity in Handball Players: A Pilot Study

**DOI:** 10.3390/ijerph192315923

**Published:** 2022-11-29

**Authors:** Gustavo García-Buendía, Darío Martínez-García, Daniel Jerez-Mayorga, Manuel Gómez-López, Ignacio Jesús Chirosa-Ríos, Luis Javier Chirosa-Ríos

**Affiliations:** 1Department Physical Education and Sports, Faculty of Sport Sciences, University of Granada, 18071 Granada, Spain; 2Strength & Conditioning Laboratory, CTS-642 Research Group, Department Physical Education and Sports, Faculty of Sport Sciences, University of Granada, 18071 Granada, Spain; 3Exercise and Rehabilitation Sciences Institute, School of Physical Therapy, Faculty of Rehabilitation Sciences, Universidad Andres Bello, Santiago 7591538, Chile; 4Department of Physical Activity and Sport, Faculty of Sports Sciences, University of Murcia, Santiago de la Ribera, 30720 Murcia, Spain

**Keywords:** internal rotation, throwing velocity, shoulder strength, dynamometry, athletic performance, handball

## Abstract

The main objective of this study was to test the relationship between shoulder internal rotation strength and standing throwing velocity. A repeated measures cross-sectional study was conducted with 20 professional handball players (mean ± SD; age: 19.28 ± 2.55 years, weight: 81.52 ± 9.66 kg, height: 185 ± 6 cm, BMI: 23.74 ± 1.69). The participants were instructed to perform eight standing throws from the 7 m line of the handball court at maximum velocity to calculate the mean and maximum throwing velocity. An incremental test was performed to calculate the repetition maximum (1-RM) of internal rotation shoulder strength. A Pearson’s correlation analysis with a 95% confidence interval (95% CI) was performed to determine whether correlations existed between dominant arm internal rotation strength characteristics and maximum and mean ball-throwing velocity. There is no correlation between the internal rotation strength of the throwing shoulder and the velocity of the ball in the standing handball throw.

## 1. Introduction

Handball physical demands involve high-intensity intermittent and vigorous effort over short periods of time with direct contact between players [1]. Athlete performance in team sports implies the development of endurance, jumping, and sprinting skills, as well as overhead throwing [2]. These physical demands depend on muscle power in order to respond to the physical and technical requirements of handball [3]. Therefore, the development of muscle strength is, along with technical and tactical skills, one of the main factors that lead to an enhancement of performance in professional players [4].

Overhead throwing is an essential skill for athletic success [4,5]. It is a complex skill that involves different body segments and joints, such as lower limb, trunk, and upper limbs [6]. During the throw, the shoulder is the central axis of the kinetic chain that generates and transfers force from the lower extremities to the ball [7]. Shoulder internal rotation and elbow extension have been pointed to as key factors for throwing performance [8]. Thus, a coordinated action of the shoulder muscles is vital for efficient performance and reduced risk of injury in handball [9,10].

Internal rotation emerges as one of the main factors involved in throwing performance [8,11]. Shoulder internal rotators benefit from the stretch-shortening cycle, directly affecting the throwing velocity in the acceleration phase [12], which leads to a high-speed movement before the ball release. Numerous studies have shown a positive correlation between strength and power in upper extremity muscles and throwing velocity [13,14,15,16]. Ortega-Becerra et al. [17] found a significant relationship between absolute load displaced at 1 m·s^−1^ in bench press and throwing velocity (*p* ≤ 0.001).

The correlation between shoulder strength and throwing velocity is currently not well established in the literature [12,18,19], although an increase in muscle strength has been shown to improve throwing velocity [20] and for implementing shoulder-strengthening programs [21]. The lack of scientific consensus on this topic might be explained by the different methods used to assess shoulder muscle strength. Several studies have measured maximum isometric internal rotational strength with a handheld dynamometer [22]. Isokinetic assessments have also been widely used to measure shoulder strength in both healthy and injured athletes [23] and is considered the gold standard method for shoulder strength evaluations [24,25,26]. Nonetheless, all of these evaluations are unrelated to the current throwing motion and the athletes’ training practice. New methods, such as elastic bands, have begun to be used for strength training and measurements [27].

Free weights training is the most widely implemented training method for all kinds of athletes. Velocity-based training has emerged as a practical way to merge both training and assessments to obtain daily results of the athletes’ strength performance [28]. This type of methodology focuses on the use of force–velocity (F-V) profiles to adjust the load on a daily basis and obtain better enhancements [29]. Functional electromechanical dynamometers (FEMD) are an alternative to perform isotonic evaluations with which to carry out an incremental test in the shoulder, in order to obtain the execution velocity at each load [30]. FEMD requires the active stabilization of participants in strength assessments, which enables a better posture for the strength application in handball-throwing sports. Previous research used FEMD in a novel way to evaluate the trunk [31,32], lower limb [33,34,35] isometric hip strength [36], muscle quality and isometric strength in elderly women [37] and in older adults with hip osteoarthritis [37,38], and shoulder rotator strength [25,39]. However, FEMD has never been used to calculate the F-V profile, and shoulder internal rotation strength has never been assessed in an isotonic way.

Thus, the main objective of this study was to explore the relationship between shoulder internal rotation strength, using the F-V profile, and standing throwing velocity in a specific population. A second objective was set to compare different variables of the F-V profile with the mean and peak velocity of handball throwing. According to previous studies, it is hypothesized that there is no correlation between the internal rotation shoulder strength and standing throw velocity.

## 2. Materials and Methods

A cross-sectional study was conducted with 30 professional handball players (mean ± SD; age: 19.28 ± 2.55 years, weight: 81.52 ± 9.66 kg, height: 185 ± 6 cm, BMI: 23.74 ± 1.69) (Table 1). The participants were informed about the research design, objectives, and risks associated with the research before giving written consent to participate. The inclusion criteria for eligible subjects were: (i) >10 years as federated handball players; (ii) not suffering, or had suffered, any shoulder injury or pain in the last 5 months; (iii) not consuming supplements or ergogenic aid during the duration of the assessments. The participants were also asked not to exercise in the 48 h before the assessments to avoid fatigue effects. The study protocol was approved by the institutional review board and was conducted in accordance with the Declaration of Helsinki.

The shoulder internal rotation strength evaluations were carried out with an FEMD (Dynasystem, Research Model, Granada, Spain) whose mechanical characteristics are an accuracy of three millimeters for displacement, a variation of 100 g when determining a load, and a sampling frequency of 1000 Hz. A self-made device was also used to fix the dominant arm with the shoulder in a 90° abducted position.

Throwing velocity was measured in a handball court with radar (Stalker sport 2, Applied Concepts Inc., Richardson, TX, USA) with an accuracy of 0.1 km-h^−1^ and an official handball size III ball (480–500 g and 56–58 cm circumference). The anthropometric data were measured with a BC-418 scale (Tanita Corporation, Tokyo, Japan) with a measurement error of 0.1 kg and HM200D digital measuring rod (Charder Electronic, Taichung City, Taiwan) with a measurement error of 0.1 cm.

The warm-up comprised a light jog for 5 min followed by dynamic stretching and the completion of sport-specific exercises such as passing or throwing at sub-maximal velocities for a total duration of 10 min. The participants were asked to perform eight standing throws at maximum velocity along the 7 m line on the handball court. Simultaneously, the evaluators stood behind the goal with the radar set up according to the manufacturer’s specifications to measure the throwing velocity. A rest of at least 10” was allowed between each throw.

After these evaluations, an incremental strength test of internal rotation was performed with an increasing load between 1 and 5 kg per set until the 1-RM was reached. Three repetitions were performed when the average velocity of repetition was greater than 0.90 m·s^−1^, two repetitions when the velocity was between 0.90 m·s^−1^ and 0.70 m·s^−1^, and one repetition when the velocity was less than 0.70 m·s^−1^. The participants’ starting position was standing with the foot opposite the throwing arm brought forward, the throwing arm in 90° abduction in the frontal plane, the elbow flexed to 90° in the sagittal plane, and the forearm in pronation (Figure 1). The cable linking the participants’ wrist to the FEMD was positioned at a 45° angulation to the participants’ forearm. This position was chosen because it most closely approximates the throwing motion [40]. The range of movement was measured registering the distance traveled by the forearm for 90° from the starting position.

After completing the incremental load test, the data were processed to obtain the internal rotation F-V profile for each player, using both average and peak velocities for each of the loads moved, following the protocols established in previous studies [41]. When obtaining the individualized F-V profiles, either with average or peak velocity measurements, the F0 and V0 variables were calculated using the equation of the slope of the straight line (Figure 2).

SPSS statistical package version 22.0 for Windows (SPSS Inc., Chicago, IL, USA) was used for statistical analyses. The descriptive data are presented as mean ± SD. The data were tested for normal distribution using a Shapiro-Wilk test. A Pearson correlation analysis with a 95% confidence interval (95% CI) was performed to determine whether correlations existed between the dominant arm internal rotation profile and maximum and average ball throwing velocity. The criteria proposed by Hopkins et al. (2009), to interpret the magnitude of the r, were null (0.00–0.09), small (0.10–0.29), moderate (0.30–0.49), large (0.50–0.69), very large (0.70–0.89), nearly perfect (0.90–0.99), and perfect (1.00). The significance level was set at α ≤ 0.05 [42].

## 3. Results

The mean and standard deviation of the mean and maximum throwing velocity achieved, as well as the descriptive characteristics of the participants who carried out the study, are shown in Table 1.

No positive correlation was observed between the participants’ internal rotation strength variables and throwing velocity (Table 2). Seven of the eight correlations are slightly negative, which represents an inverse correlation between the variables, but no significant relationship was found, either. Since the correlations are very close to 0, the magnitude of the r is null (r: 0; −0.02), small (r: −0.15; −0.15; −0.22; −0.25; −0.28), and moderate (r: −0.32), so it is impossible to determine any sense of covariation with these variables.

No correlation pattern can be seen in the graphs between the different force variables analyzed and the throwing velocity (Figure 3). Therefore, no linear relationship can be established between internal rotation strength and throwing velocity. There is no degree of agreement between the relative positions of the internal rotation strength measurements and throwing velocity. Thus, higher throwing velocity cannot be associated with higher internal rotation shoulder strength levels in this population.

## 4. Discussion

The main objective of this study was to explore the relationship between shoulder internal rotation strength, using the F-V profile, and the standing throwing velocity in handball players. To our knowledge, this is the first study to assess shoulder internal rotation strength using the F-V profile, as opposed to previous studies that assessed it isokinetically or isometrically. The results obtained in this research suggest that there is no direct correlation between these two variables, with correlations ranging from −0.28 to −0.25 for peak strength values or average strength values ranging from 0.00 to −0.22 in correlation to throwing velocity.

Although there is no clear consensus, these results are consistent with previously published studies that have found null or low correlations between handball throwing and shoulder internal rotational strength [12]. However, in recent studies, the use of force-velocity profiling or high-velocity isokinetic evaluations have found a higher correlation of the overhand throw with shoulder strength [43,44]. Small or moderate correlations were found between shoulder rotator strength and throwing velocity, regardless of the sex of the subjects analyzed in previous studies [19,43]. This contradictory phenomenon has been explained due to the sequential characteristics of the throwing kinetic chain, where velocity increases along the kinetic chain, but strength does not necessarily increase between the joints involved [45].

In addition, it must be taken into account that in the throwing pattern, the shoulder also performs adduction and flexion movements in parallel with the movement of the internal rotation; thus, these two movements can be an important factor in handball throw performance [46]. The explanation provided by previous studies is related to the fact that the kinetic chain of the handball throw is more dependent on the force generated by the legs and trunk, reducing the demand on the shoulder joint muscles to develop throwing velocity [47]. Therefore, it seems reasonable to indicate that shoulder internal rotation strength alone is not a significant variable to enhance performance in handball throwing. Other variables might be more relevant, such as intermuscular coordination or the strength of the large muscle groups related to this skill [48,49]. Although there is no clear consensus, there are studies that have correlated (r = 0.61) isokinetic ER shoulder strength (240°·s^−1^) with standing throwing speed in adolescent handball players [43]. Nevertheless, this same study emphasizes that the main role of the shoulder rotators is to stabilize the joint. The main role of shoulder rotator muscle training should be focused on gaining stability for enhanced force transmission through the overall action [50].

Reviewing the literature, it is possible to find that not only the strength value itself has been taken into account when attempting to explain the moderate correlation obtained between both variables. Neuromuscular coordination has been pointed out as an important factor to consider [46,51]. Other variables, such as experience level and throwing automation or power have also been taken into account to explain these differences [52]. In this regard, it has been found that novice players do not reach their angular velocity peak of internal rotation until releasing the ball [53]. Differences between multiple handball throws have been analyzed (standing, running, or jumping), obtaining similar results in all modalities [54].

However, these results should be interpreted with caution because the sample is not large enough to generalize the findings. Furthermore, it is difficult to compare the isolated shoulder internal rotation movement, despite establishing evaluation conditions that are more similar to the throwing gesture and the type of contraction produced with the physical demands of a match. It is not clear which are the most relevant factors in predicting throwing performance. Age, body mass, skill level (amateur or professional), throwing technique (supporting, three-step throwing, jump throwing, with or without target) or the interaction between several of them may play a key role. In addition, this research performs isotonic internal rotation measurements, setting the F-V profile for this movement, being the first study to compare this type of movement with throwing velocity. Future lines of research should test whether these results would also occur in female players.

## 5. Conclusions

The current findings have significant practical implications for handball players. Throwing velocity was not related to internal rotation shoulder strength. Consequently, coaches and clinicians are advised to measure other variables rather than shoulder internal rotation in isolation as an indicator of throwing performance. In addition, they are encouraged to seek to increase the strength and power of the handball throw in a holistic and not segmented manner. Because of these results, our hypothesis is that the function of shoulder rotators is to stabilize the joint. Therefore, clinicians should continue to evaluate shoulder internal rotation for injury status or injury risk, but not as a performance or return-to-play indicator.

## Figures and Tables

**Figure 1 ijerph-19-15923-f001:**
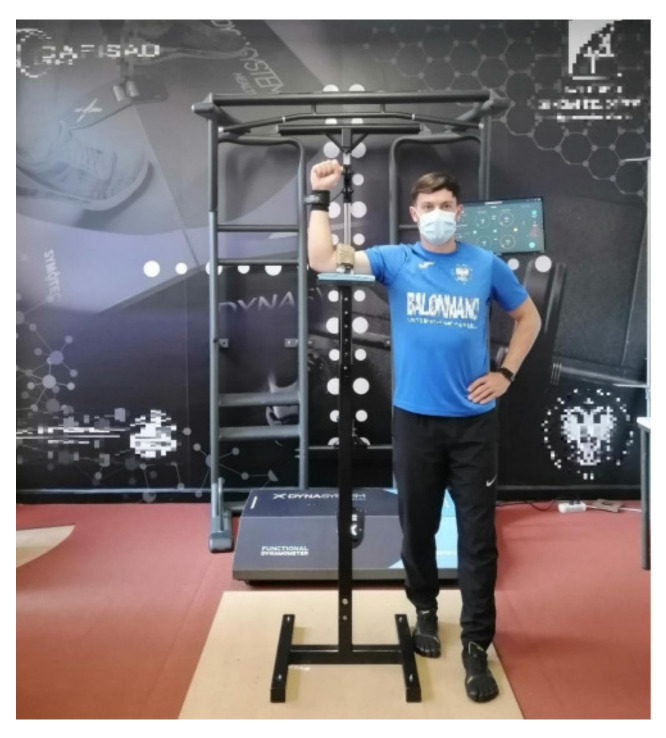
Shoulder internal rotation strength measurement.

**Figure 2 ijerph-19-15923-f002:**
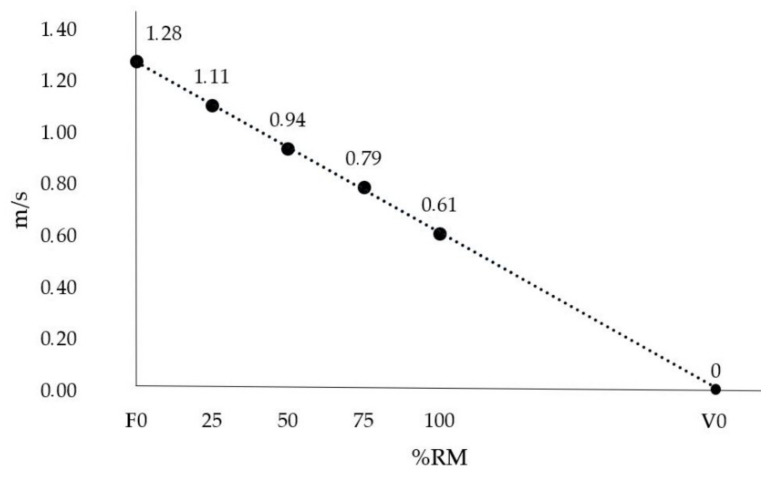
Shoulder internal rotation’s F-V profile.

**Figure 3 ijerph-19-15923-f003:**
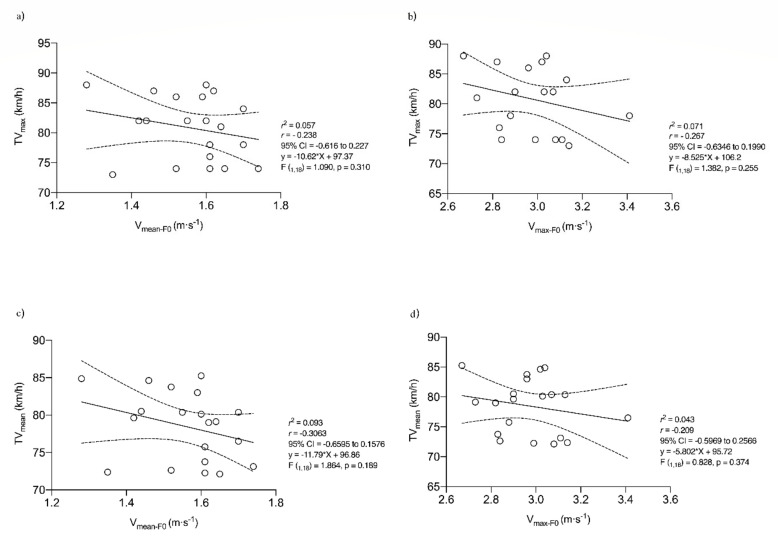
Linear regression between mean (TVmean) and maximum (TVmax) throwing velocity and value where the force–mean velocity profile curve intersects the velocity axis (Vmean-F0) and value where the force–maximum velocity profile curve intersects the velocity axis (Vmax-F0). (**a**) Linear regression between TVmax and Vmean-F0. (**b**) Linear regression between TVmax and Vmax-F0. (**c**) Linear regression between TVmean and Vmean-F0. (**d**) Linear regression TVmean and Vmax-F0. (**e**) Linear regression between TVmax and Fmean-V0. (**f**) Linear regression between TVmax and Fmax-V0. (**g**) Linear regression between TVmean and Fmean-V0. (**h**) Linear regression between TVmean and Fmax-V0.

**Table 1 ijerph-19-15923-t001:** Descriptive characteristics of participants.

	Age (Years)	Weight (kg)	Height (cm)	BMI	TV Mean (km·h^−1^)	TV Max (km·h^−1^)	Vmean-F0 (m·s^−1^)	Fmean-V0 (N)	Vmax-F0 (m·s^−1^)	Fmax-V0 (N)
Mean	19.28	81.52	185	23.7	78.45	80.8	1.56	169.51	2.97	181.44
SD	2.55	9.66	6	1.69	46	5.3	0.119	24.44	0.16	34.81

Abbreviations: TV mean: average velocity of the three best throws; TV max: maximum velocity achieved in the best throw; Vmean-F0: value where the force–mean velocity profile curve intersects the velocity axis; Fmean-V0: value where the mean force–velocity profile curve intersects the force axis; Vmax-F0: value where the force–maximum velocity profile curve intersects the velocity axis; Fmax-V0: value where the maximum force–velocity profile curve intersects the force axis.

**Table 2 ijerph-19-15923-t002:** Correlations between different force variables and average and peak throwing velocities.

Parameters		ES	Pearson Correlation	95% CI Lower–Upper	ICC	*p*-Value
Mean Throwing Velocity	Vmean–F0	21.11	−0.25	−0.64–0.25	−0.07	5.86
	Fmean–V0	5.76	−0.15	−0.56–0.31	−0.07	2.99
	Vmax–F0	20.73	−0.28	−0.64–0.19	−0.02	8.75
	Fmax–V0	−3.45	0.00	−0.63–0.63	−0.26	0.0001
Max Throwing Velocity	Vmean–F0	23.64	−0.32	−0.68–0.17	0.05	7.22
	Fmean–V0	1.62	−0.15	−0.55–0.32	−0.06	−5.93
	Vmax–F0	23.19	−0.22	−0.60–0.25	−0.02	1.02
	Fmax–V0	−3.67	−0.02	−0.45–0.43	0.00	4.76

Abbreviations: ES: effect size; CI: confidence interval; ICC: intraclass correlation; Vmean-F0: value where the force–mean velocity profile curve intersects the velocity axis; Fmean-V0: value where the mean force–velocity profile curve intersects the force axis; Vmax-F0: value where the force–maximum velocity profile curve intersects the velocity axis; Fmax-V0: value where the maximum force–velocity profile curve intersects the force axis.

## Data Availability

Data can be made fully available upon request.

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
