# Peer review of "Exploration of the Shoulder Internal Rotation’s Influence on Throwing Velocity in Handball Players: A Pilot Study"

_ijerph, 2022, doi:10.3390/ijerph192315923_

Round 1

Reviewer 1 Report

1) It is better for the term internal rotation to be mentioned in full throughout the paper rather than supplementing it with the abbreviation IR, for improved readibility.

2) The statement "Nowadays, correlation between shoulder strength and throwing velocity is not well established in the literature [18,19]." only uses references that are dated, namely from 2014 and 2016. In addition, one of the references focuses only on women players. Perhaps more references from recent years are required to back this statement.

3) Line 209, the word "analyzed" is misspelled as "analized".

4) Line 211, on the statement "However, these results should be interpreted with caution because of the sample is not enough to generalise these findings.", I too advise the authors to administer these words with caution. Perhaps a modification of the title and objective to indicate that this study was indeed a preliminary study is required in order for such a statement to be justified. If not, parts of the discussion might appear to be "overclaimed" to other researchers.

5) Add more references from the years 2020-2022.

Thank you.

Reviewer 2 Report

This study examines the correlations between shoulder internal rotation (IR) strength and standing throwing velocity. However, throwing is a holistic technique and the citation [34] has suggested IR was not a good indicator of throwing velocity at any type of throw. Did author make a new different on and theory or technique?  It's a major issue in the Methods that should be addressed. 

Line 76: Participants were training together? If not, how you prove the throwing skill and force technique of shoulder was consisting. And the throwing performance will be affected by playing position?

Line 104-106: Any reason or citation support the setting?

Line 107: Describe the standing posture (parallel feet or front and back) Whether in penalty shot conditions?

Method: Description of the experimental procedure is insufficient.

Table 1: confirm the value of TVmean and TVmax, one maybe wrong.

Table 1: Please add a figure to describe the velocity and force curve intersects.

Table 1: I am confused why Fmean-V0 was large than Fmax-V0?

Reviewer 3 Report

As you mentioned in your manuscript "These results support previous studies such as [37] where no correlation was found 180 between isokinetic IR strength"

Why are you still investigating this phenomenon without any other modification/ to the overall design? 

Reviewer 4 Report

GENERAL & SPECIFIC COMMENT:

Thank you for the opportunity to review this interesting issue and we value the effort that you put into this study. I think that this research is able to get the reader interested and work together on the topic. I think this article has some potential but there are some critical flaws as described below.

1. Abstract

-Appropriate, however, do you expect the same results for other overhead throwers such as baseball and tennis players, except handball players?

You must indicate that you are in handball players on the conclusion of Lines 16 and 221

2. Introduction

-Appropriate, however, (2018) in Line 50

3. Methods

-Line 90, your study measured IR strength and throwing velocity at the throwing position. How was the ROM set up? In most of the throwing athletes, the external rotation ROM is large, resulting in greater IR torque and velocity can be implemented.

- The throwing velocity mentioned in your study can be measured in the acceleration phase and it is an important point. In this study, how did you implement the acceleration phase to evaluate the throwing velocity in the posture shown in figure 1?

4. Results

- Line 154, table 4?

5. Discussion

- Line 193, in the results of this study, IR strength is not related to throwing velocity. In particular, in the discussion section, kinetic chain was mentioned. If so, wouldn't IR strength training alone in the overhead throwers be effective in improving the throwing velocity? Does external rotators strength matter? In most of the throwing athletes, the external rotation ROM is large, resulting in greater IR torque and velocity can be implemented.

- Line 224, shoulder internal rotation -> IR

6. Conclusion

-Appropriate, however, do you expect the same results for other overhead throwers such as baseball and tennis players, except handball players?

You must indicate that you are in handball players in the Line 221

7. References

-Appropriate

8. Figure

-Appropriate

9. Table

-Appropriate, however, table 4? and Table 2 ->95% CI

Round 2

Reviewer 2 Report

The authors stated that there is no correlation between the internal rotation strength of the throwing shoulder, with the velocity of the ball in the standing handball throw. The gap in the current research is not being able to distinguish the emphasis of throwing and shoulder internal rotation, how important its plays in handball player. However, the purpose of this study (explore the relationship between shoulder internal rotation strength and standing throwing velocity) and hypothesis doesn’t seem to explain the importance of present research for handball athletes. They sound very like previous studies the authors mentioned. The authors need to re-organization the introduction to make it a stronger statement on what’s new/novel about this manuscript.

The discussion is not very convincing, especially with the unclear statement in the Introduction I mentioned above. By mentioning a lot of consistency with previous studies, it makes me feel even less novelty in this study.

Some IR were not revise, (line 46, 203 etc.) and 42 need IR?

Line 239: Please specify what practical implications.
